# Non-Thermal Atmospheric Pressure Plasma Application in Endodontics

**DOI:** 10.3390/biomedicines11051401

**Published:** 2023-05-09

**Authors:** Ana Bessa Muniz, Mariana Raquel da Cruz Vegian, Lady Daiane Pereira Leite, Diego Morais da Silva, Noala Vicensoto Moreira Milhan, Konstantin Georgiev Kostov, Cristiane Yumi Koga-Ito

**Affiliations:** 1Department of Environment Engineering and Sciences Applied to Oral Health Graduate Program, São José dos Campos Institute of Science and Technology, São Paulo State University (UNESP), São José dos Campos 12247-016, SP, Brazil; bessa.muniz@unesp.br (A.B.M.);; 2Department of Physics, Faculty of Engineering in Guaratinguetá, São Paulo State University (UNESP), Guaratinguetá 12516-410, SP, Brazil

**Keywords:** non-thermal plasma, endodontics, root canal, microorganism

## Abstract

The failure of endodontic treatment is frequently associated with the presence of remaining microorganisms, mainly due to the difficulty of eliminating the biofilm and the limitation of conventional irrigation solutions. Non-thermal atmospheric pressure plasma (NTPP) has been suggested for many applications in the medical field and can be applied directly to biological surfaces or indirectly through activated liquids. This literature review aims to evaluate the potential of NTPP application in Endodontics. A search in the databases Lilacs, Pubmed, and Ebsco was performed. Seventeen manuscripts published between 2007 and 2022 that followed our established inclusion criteria were found. The selected manuscripts evaluated the use of NTPP regarding its antimicrobial activity, in the direct exposure and indirect method, i.e., plasma-activated liquid. Of these, 15 used direct exposure. Different parameters, such as working gas and distance from the apparatus to the substrate, were evaluated in vitro and ex vivo. NTPP showed a disinfection property against important endodontic microorganisms, mainly *Enterococcus faecalis* and *Candida albicans.* The antimicrobial potential was dependent on plasma exposure time, with the highest antimicrobial effects over eight minutes of exposure. Interestingly, the association of NTPP and conventional antimicrobial solutions, in general, was shown to be more effective than both treatments separately. This association showed antimicrobial results with a short plasma exposure time, what could be interesting in clinical practice. However, considering the lack of standardization of the direct exposure parameters and few studies about plasma-activated liquids, more studies in the area for endodontic purposes are still required.

## 1. Introduction

Microbial presence is the primary etiological cause of pulp and periapical infections [1]. Root canal infections are mediated by microbial species that form biofilms. Multispecies composition combined with the radicular system complexity makes the canal system disinfection a complex procedure. In this context, microbial persistence is the factor that mostly contributes to failures in endodontic treatment. It allows root canal infection recidivism. This reality negatively affects the quality of life of the patients [2]. Among the several agents involved in persistent endodontic infections are *Enterococcus faecalis* and *Candida albicans.* The pathogenicity of these microorganisms is related to their ability to survive in the dentine tubules for an extended period, keeping them viable even in scarcity [3].

Endodontic treatment success primarily depends on the capacity to eliminate the intracanal infection source [4]. Therefore, root canal disinfection represents a necessary procedure before canal filling. It includes conventional mechanical–chemist methods, such as debridement and irrigation with a chemical agent.

The use of non-thermal atmospheric pressure plasmas (NTPPs) in the biomedical sciences has emerged as a promising alternative in several areas. Their properties, mostly related to the generation of reactive oxygen and nitrogen species (RONS) [5], allow microbial deactivation, making NTPPs advantageous for the sterilization of implants and medical/dental instruments [6,7,8]. In addition to the antimicrobial properties, NTPPs have demonstrated anti-inflammatory effects, favoring tissue repair [9,10]. Different works focused on cancer treatment have also investigated NTPPs as a promising tool to act as an adjuvant therapy against cancer [11,12]. In addition, antifungal, antibacterial, antiparasitic and antiviral properties have been described [13,14].

Cells, tissues or materials can be exposed to NTPPs in two different ways: direct, when NTPP in the form of a plasma plume is applied to a given substrate, or indirect, when a liquid is activated by an NTPP prior to its application to the substrate [15]. Both direct and indirect methods have shown good perspectives in dentistry [16]. Since root canal reinfection is recurrent and the conventional methods and techniques have failures and risks [17,18], the use of NTPPs seems to represent an alternative to promote disinfection. Therefore, this review aims to discuss the main problems related to endodontic failures and the possible role of NTPPs in this context. For this, the main advances in using NTPPs in the endodontics field will be presented and discussed.

## 2. Design of the Study

This review is divided into six sections. The third section will introduce the possible role of NTPPs in the endodontic field. For this, the microbiota related to endodontic infection, the conventional irrigants used in the endodontic treatment, as well as their limitations, and the main physicochemical properties of plasma will be discussed. Section 4 will explore the main text of this integrative literature review about NTPPs applied in the endodontic field, followed by a discussion in Section 5 and finally the conclusion of the work in Section 6.

In this integrative literature review, recent studies focused on the effect of direct and indirect NTPPs exposure on microorganisms of endodontic interest were selected. The following keywords, Nonthermal Plasma, Endodontics, Cold plasma and Treatment, were used in four databases (Science Direct, Lilacs, Pubmed, and Ebsco). In the general search, 279 reviews were found. From these, only research studies or literature reviews that investigated NTPPs applied to the endodontic area from 2007 to 2022 were included. Reference lists of referral studies were inspected to identify any additional relevant published data. Studies investigating the application of non-thermal atmospheric pressure plasmas in other areas of dentistry were not considered. After the evaluation of the mentioned criteria, 17 papers that included in vitro and ex vivo studies on the use of NTPPs applied to the endodontic area were selected.

## 3. Possible Role of NTPPs in the Endodontic Area

### 3.1. Infectious Endodontic Microbiota

The main cause of endodontic failure is the presence of microorganisms that lead to intra-rooted and extra-rooted infections and become resistant to disinfection protocols [19]. Although fungi, archaea and viruses contribute to the microbial diversity in endodontic infections, bacteria are the most common microorganisms present in these infections. Their diversity varies significantly according to the infection type. Moreover, different penetration levels in the endodontic invasion space can be observed in different conditions, such as caries lesions, traumatic pulp exposure, and fractures [20,21].

Root canal bacteria can be isolated as planktonic cells, suspended in the root canal liquid phase. However, aggregates or congregations can adhere to the root canal walls, forming biofilm layers. Biofilms are the bacteria growing model, where the sessile cells interact to form dynamic communities linked to a solid substrate. They are found in an extracellular polymeric matrix [22]. Endodontic infection can be classified as primary and secondary, wherein the first usually presents a great wealth of species. In contrast, in the second category, there are one or two species [2].

The bacteria frequently associated with persistent endodontic infection after treatment are usually Gram-positive. There is an equal distribution of facultative and obligate anaerobic microorganisms, with a prevalence of streptococci and enterococci [21]. Other bacterial species have been related to endodontic infections. The first endodontic infection clinic case with *Klebsiella variicola* has recently been reported [22].

The most common species involved in apical reinfection is *Enterococcus faecalis*, which is responsible for 80% of endodontic infections in humans. Its pathogenicity can be attributed to its high survival ability inside dentinal tubules [4,21,23]. Its environmental resistance is due to characteristics such as the species’ capacity to form biofilms and ability to support high pH, such as the one used in intracanal medication with Ca (OH)_2_ [24]. This pathogen tends to stand out when in a stable multi-species microbiota, becoming the dominant species [21]. It is estimated that the *E. faecalis* biofilm is about 1000 times more resistant to antimicrobial agents when compared to its planktonic counterpart [1].

For decades, endodontics microbiologists concentrated their studies on the role of bacteria in the etiopathogenesis of endodontic infection. Nevertheless, in recent years, research in this field has demonstrated an emerging interest in microorganisms from other kingdoms, such as archaea, viruses and fungi [25]. In the Fungi kingdom, *Candida* species are the most frequently related to endodontic reinfection. This yeast is present in a low proportion in endodontic infection (around 10%). However, *Candida albicans*, which is the species most frequently detected, presents a low susceptibility to the intracanal disinfectants commonly used, being able to survive inside the dentinal tubules due to the ability to form biofilms, leading to reinfection [23,25,26]. Its pathogenicity evolves dental pulp and peri-radicular tissue cells, mainly in immunocompromised patients [26].

### 3.2. Conventional Endodontic Irrigants

The endodontic therapy approach aims to reduce the microbial load to a subclinical level, removing the necrotic material and disinfecting the canal through the chemical–mechanical preparation using lime instrumentation, disinfection with irrigants, and subsequent intracanal medication [2].

Disinfection solutions, such as sodium hypochlorite (NaOCl) or chlorhexidine, and chelating (i.e., with EDTA) show outstanding performance during disinfection [17]. Although highly effective, NaOCl solutions have limited penetration capacity in the apical part and the dentinal tubules, mainly in bi- or multi-rooted teeth with more complex intra-rooted anatomy [4].

For better efficacy, techniques such as applying ultrasound to reduce the surface tension or the pressure increment in the irrigation moment have been proposed. However, during this procedure, fluid extravasation can injure the healthy periapical [4,18]. This is related to the fact that sodium hypochlorite is highly toxic and can lead to an inflammatory process in the vital tissues, leading to tissue necrosis. It occurs due to the high sodium hypochlorite oxidative potential. Furthermore, NaOCl can modify dentin properties and characteristics, such as microhardness and resistance. These changes can affect the stability of the dental element [17,18].

In this sense, an innovative non-thermal atmospheric plasma jet technique can represent an alternative to the conventional treatment [18]. Studies have reported the efficacy of a plasma jet against biofilms inside the root canal, which suggest that this technology can be helpful as an adjuvant in conventional disinfection therapies, seeking to decrease the discussed limitations. One of the advantages of plasma is its capacity to reach microorganisms lodged in narrow niches, in addition to low toxicity to eukaryotic cells [23].

### 3.3. Main Physicochemical Properties Related to NTPPs

Plasma is a complex mixture of electrons, ions and neutral species and is generally considered to be the fourth state of matter. Plasma is abundant in nature (stars, auroras, lightning, etc.) and can also be produced in the laboratory at different conditions using electrical discharges. Many plasma devices, such as plasma torches and microwave discharges, are operated at high temperatures. It is also possible to generate non-thermal plasma in which the gas temperature is quite low. Especially attractive for applications is the non-thermal plasma produced at atmospheric pressure, because it does not require the use of vacuum systems [27]. NTPPs have two crucial features that make them applicable in medical and dentistry fields: plasma has high plasma has biological effects due to the reactive species, and due to the reactive species, and it does not damage the tissue structure [4].

Usually, non-thermal plasmas are generated by applying short electrical pulses to a discharge gap, where through gas ionization unstable reactive species are generated while the temperature is kept stable [27]. NTPPs are obtained from a low-temperature gas (<40 °C), which does not induce thermal injuries to the tissues [17]. These gases are partially ionized, containing highly reactive particles such as excited atoms, oxygen and nitrogen reactive species, U.V. irradiation, photons and electrons [1,2,28]. The main reactive oxygen species (ROS) include ozone, atomic oxygen, superoxide, hydrogen peroxide and hydroxyl radicals [29]. ROS present key roles in cell signaling pathways and redox reactions, with potential for biomedical applications [30].

Gases such as Helium (He), Argon (Ar), Nitrogen (N), Oxygen (O), atmospheric air and a mixture of the previously cited ones are frequently used for NTPP generation. NTPPs compounded by Ar/O_2_ and Ar/air have been considered the most efficient [23].

## 4. NTPPs Applied to Endodontics

The aim of root canal treatment is the reduction in viable bacteria in the local. Bacterial persistence during canal filling is one of the risk factors that can lead to apical periodontitis post-treatment [20]. The application of NTPPs has demonstrated promising results in association with conventional endodontic treatment regarding their antimicrobial effects [23]. There are two modalities of treatment that have been explored in vivo, in vitro and ex vivo: direct and indirect. While in the direct modality the area of interest on the substrate is directly exposed to NTPPs, in the indirect one different liquids which have been previously activated by plasma are then applied to the substrates. The indirect plasma application is based on the fact that when plasma is in contact with liquid its reactive species interact with the liquid molecules, producing other long-lasting reactive species in the liquid. The plasma-activated liquids’ PALs can be frozen and stored for long periods of time before being used for biomedical treatments. The available literature on the use of NTPPs in Endodontics (Table 1) will be discussed in different subtopics.

### 4.1. Direct Method of NTPP Application

One of the first ex vivo studies using NTPP inside a root canal with a microbial biofilm from saliva showed that the application of plasma for 5 min reduced the total amount of microorganisms until 1 mm deep into the dentin root canal when compared to the non-treated canal. The NTPP was generated from a mixture of He and O_2_ (1%). Additionally, the authors performed an in vitro test with *Bacillus atrophaeus* that showed 100% disinfection. The authors mentioned, in addition, that the biofilm removal level and the dispersion level inside the root canal differed depending on the treatment parameters [31].

To evaluate *E. faecalis*’ inactivation by NTPP, Pan et al. [3] submitted 85 single-rooted teeth to treatment with argon and oxygen plasma (98% + 2%) with a gas flux of 5 L/min for periods varying between 2 and 10 min. After, they evaluated the viability of bacteria cells with Scanning Electron Microscopy (SEM) and Laser microscopy. The authors found that the inactivation level increased over the plasma exposure time. In this scenario, the periods between 8 and 10 min were the most efficient in achieving reliable disinfection. The SEM images showed the biofilm rupture in the tubules and on the dentin surface after 10 min of NTPP exposure.

Some investigations using NTPP on the most prevalent species in endodontic infection have demonstrated promising results in the disinfection process. A reduction in *E. faecalis* planktonic cells was observed after He-generated NTPP exposure for 2 min [32].

The Plasma Creator device model RC-2 has been used to study plasma application with chlorhexidine. For this, a gas tube containing He and O_2_ (1%) and chlorhexidine (2% solution) was connected to the device. To test this model, 120 bovine dentin disks were experimentally infected with *E. faecalis* or microcosm biofilm collected from a clinical infection. The results indicated that the plasma associated with chlorhexidine was more effective than the isolated treatment. It showed an 80% reduction in bacterial viability when applied for 5 min [33].

Ledernez et al. [18] evaluated the in vitro NTPP effect against microorganisms of medical interest, such as *E. faecalis*, *S. mutants*, *S. aureus*, *P. aeruginosa* and *E. coli*. The working gas was a mixture of He and O_2_ with a flux of 1 L/min. The exposure was performed for 3 min at 1–3 mm from the Petri dish surface. It was observed that the disinfection efficacy varied according to the species. *E. faecalis* was the most resistant, while *E. coli* was the most susceptible to the plasma treatment. It was also found that the halo area (0.25 mm^2^) corresponding to the decontamination was higher than the target area in the root treatment.

A recent study evaluated the effect of a gas mixture of He and O_2_ plasma applied on ex vivo *E. faecalis* biofilms. These biofilms were grown inside single-root teeth. Thus, Saleewong et al. [24] evaluated the plasma effect with a gas flux of 0.5 L/min in 54 teeth. The dental elements were allocated to five experimental groups (NaOCl, NTPP, NaOCl + NTPP, only the gas, and the negative control). The results demonstrated a significant reduction when using the NTPP for 5 min, or for 1 min in association with NaOCl (5%). Moreover, the association of both decontamination techniques led to a greater depth level of decontamination in the dentin.

To evaluate the treatment with NTPP in depth dentinal tubules, an NTPP generated with dielectric barrier discharge (DBD) was applied at 1 mm from the prepared root. The human teeth had a length of 14 mm, and the treatment was performed for 5 min [35]. The authors verified that NTPP had a disinfection capacity almost similar to NaOCl (2.5%) and seemed to be more effective in the middle part of the dentin. Minor effects were observed in the coronal and apical thirds.

Other investigations about NTPP antimicrobial effects in different depths of root dentine were performed. Herbst et al. [37] used argon plasma for 60 s isolated or in association with CHX (2%). The exposition time was 30 s inside roots from human premolar teeth. They observed that the combination of CHX and NTPP was more effective than isolated treatments in the general disinfection, inside the coronary part (0–300 µm) and also in the deeper layers (500–800 µm). These results suggest that an antimicrobial reduction to a maximum depth of 800 µm can be reached by using NTPP as an adjuvant.

The comparison of NTPP and other antimicrobial agents was also evaluated in a study with infection root canal simulation. For this, the teeth were infected with *E. faecalis* cells. Ex vivo treatments with Ar NTPP, NaOCl (0.6%) or CHX (0.1%) were performed for 3 min [34]. The treatment with NTPP had visible destructive effects and was better at the elimination of the residual bacteria compared to the control and the conventional irrigants. However, the visible destruction of the cellular walls and the most effective reduction in the microbial load were observed with the association of the treatments. In this study, the disinfection action of NTPP could be considered superior to the CHX and equal to NaOCl 0.6%.

Another study compared ex vivo teeth treated with the He/O_2_ NTPP for 30 min with teeth treated with conventional irrigants. Microorganism reduction was observed in both cases. According to the authors, the concentration of the irrigants and the application time of the irrigants and plasma are key factors. For example, treatments with NaOCl at 6% were shown to be 4x more potent when compared to the He and O_2_ (1%) NTPP, both applied for 30 min in each extracted human teeth root canal [27].

Armand et al. [1] submitted 60 single-rooted human teeth to He and O_2_ combined NTPP under 4 L/min with an injector nozzle of 2 mm. The injector nozzle was inserted in the root canal for a time between 4 and 8 min. The results were compared to the control group, and the group treated only with He. All NTPP groups significantly reduced the microbial load, but the mixed gas worked better than the pure He NTPP.

Interestingly, a study showed that the treatment with Ar/O_2_ (2%) NTPP on the root canal reduced 98.8% of *E. faecalis* colonies after 8 min of exposure. When the microbial population was evaluated after seven days of the treatment, a reinfection was detected. The authors related that it would be necessary to have 30 min of NTPP exposure to prevent the reinfection [38]. Nevertheless, NTPP was shown to be a fast and efficient alternative when compared to the traditional treatment composed of intracanal medicine, which could stand for 1 or 2 weeks. Another study where NTAPP was operated with the same gases verified that treatment for 12 min eliminated three-week *E. coli* biofilms from the root canal [28].

Resistant microorganisms commonly seen in oral infections are not restricted to bacterial species, but also appear in fungi. Between them, some strains of *C. albicans* can lead to a persistent infection process. It is estimated that fungal infection can be present in approximately one in every ten endodontics infections. From these, 9% are observed in primary disorders and 9.3% in secondary ones. However, the role of fungi in the pathogenesis of endodontic infection is still unclear [39].

In this sense, Doria et al. [36] investigated the effect of NTPP using a mixture of Ar and air on *C. albicans* biofilm. The authors worked with three groups: control, NTPP and the plasma air mixture. The microorganism reduction was 85% and 88.1% in the plasma and air plasma groups, respectively. The fungal viability was 33% in the plasma group and only 8% in the association group (air plasma), demonstrating that this technology could represent an exciting alternative for medical-dentistry applications.

Kerlikowski et al. [23] evaluated Ar/air NTPP in the proportion 99:1 with a flux of 5 L/min. The exposure was performed in incisive maxillary and mandibular premolars infected with *C. albicans.* The study compared the effect of pure gas, saline solution (0.9%), CHX at 2%, NaOCl at 5.25%, octenidine (0.1%) and Ar/O_2_ NTPP. The most efficient treatments were single NTPP, followed by the treatments where plasma was associated with a common irrigant solution. The treatment consisted of irrigant contact for 6 min and NTPP exposure for 6 more min. When considering monotherapies, the application of plasma for 12 min stands out compared to the conventional irrigants.

### 4.2. Indirect Method of NTPP Application

To investigate He NTPP for endodontic disinfection, some authors used a handheld Plasma Gun (PG) [17]. The authors evaluated the direct and indirect modalities of treatment. For the indirect procedure, distilled water was activated, generating plasma-activated water (PAW). PAW was prepared through the exposure of 100 µL of sterile water to NTPP for 1, 3 and 5 min in 96-well-plates and immediately used in the root canals. The root canals were contaminated by *E. faecalis* in a wet and dry environment. They verified a higher microbial load reduction in a humid climate when the PAW was activated for 5 min. In the same study, it was observed that NTPP direct treatment in a dry environment showed the best decontamination results among all the procedures, including 3 min of NaOCl (0.6%) and CHX (0.2%).

Another recent study using PAW activated with compressed air on *E. faecalis* verified that there was a planktonic cell reduction with treatment for at least 45 s. A decreased capacity of the microorganisms to grow biofilm and to express virulence was also observed after the treatment [28].

## 5. Discussion

NTPP is attractive because it combines conventional treatment efficiency and outstanding safety. Different generation sources may be related to NTPP [17]. The direct application was demonstrated to be heterogeneous, mainly due the differences between the used parameters. These parameters include time of exposure, work gas, and power supply characteristics [1].

The works analyzed in this study showed a large work gas variety. The most used work gas was pure helium [17,32] or its association with oxygen [1,17,18,24,27,31,33]. The use of argon was described in seven studies [3,23,28,34,36,37]. Only one study evaluated the exclusive usage of ozone to generate NTPP [35].

The exact mechanism that promotes the microorganism inactivation and destruction is still unclear. It is assumed that U.V. irradiation and charged particles promote microorganism disruption through atomic oxygen or irradiation, damaging the extracellular biofilm matrix, which is compounded by a polymer [3]. Reactive species and atoms obtained from molecular oxygen are essential due to their capacity to penetrate the cells [1]. However, a recent study refutes the theory that plasma can act in cell wall destruction after spectrophotometric analysis did not find any intracellular liquid extravasation clues, such as nucleic acids [40].

Preliminary studies about the effect of PAW on *E. faecalis* planktonic cells [28] and *C. albicans* [41] showed an inhibitory activity of these microorganisms with a decreased expression of virulence genes [28] In this way, PAW is also a promising adjunct treatment for the endodontic area, as well as what it has shown in other dentistry areas [42].

To our knowledge, plasma and plasma-activated liquids present different mechanisms of action. In both cases, the bactericidal effects can be attributed to a combination of physical-chemical products. Direct exposure is mainly related to the formation of ROS and U.V., while RNS has been observed in higher amounts in PAWs. RNS and ROS can damage cellular components such as the cellular walls, proteins, and genetic material. However, the solubility effect of reactive species found in PAW differs from species generated in the gas phase. Moreover, environmental interactions can be observed in PAW because changes may happen to composition and efficacy during the treatment time [16,42,43]. It is worth mentioning that RNS are typical in aerobic environments, so the bacteria may preset the natural protection system against such aggression, represented by enzymes, molecules, and neutralized proteins [43].

Saleewong et al. [24] pointed to a high penetrability ability of NTPP in the gas phase to reach the microorganism after the irrigant usage. This study also showed that without previous electric or electromagnetic activation to ionize the gas, He exposure did not show a bactericidal effect. The reactive species generated in NTPP, such as atomic oxygen, ozone, and hydroxyl radicals, seemed to confer disinfectant capacity.

In the reviewed studies, the minimal exposure time was 2 min [3,32,38] while the most frequent application time was equal to or superior to 5 min [23,24,31,35]. The maximum application time was 30 min, as Schaudin et al. [27] described. The researchers assigned a direct relationship between the exposure time and the disinfection capacity. According to Bansode et al. [32], an exposure time of 2 min was enough to reduce the microbial load significantly. On the other hand, Du et al. [33] estimated that after 2 min the microbial load could be reduced by 54%, while a reduction of 80% could be reached after 5 min, when compared to CHX.

On the other hand, several studies have demonstrated that the ideal time of exposure is greater than 8 min [3,28,38]. Pan et al. [3] noticed that the microbial inactivation increased with exposure time, with the period between 8 and 10 min being the most efficient for disinfection, with biofilm rupture in the dentine surface tubules and a reduction in biofilm viability after 10 min.

Li et al. [28] evidenced the destruction of biofilm architecture and the decrease in its thickness after NTPP exposure, where plasma achieved the ideal intra-root disinfection after 12 min. Wang et al. [38] obtained great disinfection results at 6–8 min of Ar/O_2_ NTPP. The authors observed that the disinfection through the root was heterogeneous, so in the apical region it was still possible to find considerable viable amounts of microorganisms. Because of this, they concluded that the best result to avoid reinfection would be NTPP exposure for 30 min, which could lead the reinfection risk to zero.

According to Ledernez et al. [18], there are differences in the microorganism susceptibility to NTPP action, i.e., the antimicrobial effect of NTPP may change according to the microbial species. *E. faecalis* has been presented as the most resistant, while *E. coli* has been considered the most susceptible to NTPP treatment. *Streptococcus* and *Staphylococcus* spp. have shown intermediate susceptibility. NTPP may be efficient on young and mature *E. faecalis* forms [1]. The vulnerability can be attributed to different compositions of the cell wall. Usually, Gram-positive bacteria have a thicker polyglycan layer when compared to Gram-negative bacteria, which can act as a shield against NTPP action [43].

The tooth canal anatomy can be complex. Moreover, most of the research focused on NTPP to the endodontic area until the present has been based on ex vivo models using single-rooted teeth. Armand et al. [1] warned that this dental type is less complex and has a straighter canal compared to multichannel teeth. Therefore, the NTPP effect on multichannel teeth may present a different behavior.

The surface complexity on which the treatment is performed must be considered once the plane surface does not represent the canal cavity structure and can show outstanding results in a short treatment time [23].

According to Schaudinn et al. [27], NTPP is, in theory, able to reach the whole canal extension because the reachable area is superior to the intra-root surface. Using devices such as applicator nozzles, it is estimated that the decontamination area corresponds to five times the nozzle area and reaches a size equivalent to 0.25 mm^2^, which could promote enough space to disinfect the walls of the whole tooth canal [18].

However, NTPP efficacy has been shown to be heterogeneous through the canal. Üreyen Kaya et al. [35] highlighted the high NTPP ability to eliminate microorganisms on dentinal tubules and canal walls. Despite this, the same could not be observed in the coronal and apical areas. According to the authors, the highest disinfection capacity could be attributed to the highest NTPP formation under the nozzle at 5–6 mm. This area corresponds to the medial region.

NTPP’s disinfectant effect could be potentialized by associating two or more techniques. As presented in the other section, Herbst et al. [37] showed that in all treatments a significant reduction in the microbial load compared to the control and the highest decrease occurred in the combination of CHX and NTPP at 0–300 µm. There was a low disinfection capacity in depth between 500 and 800 µm when combining treatments compared to a single treatment. The authors attributed this result to the root canal anatomic features. On the other hand, Saleewong et al. [24] observed that NTPP association with a common disinfectant (NaOCl) could increase the efficacy, mainly in deeper dentin levels.

It is essential to highlight that NaOCl in high concentrations, as Schaudinn et al. [27] used in their study, could not be clinically viable. A high irrigant concentration and high time of application can provide the best antimicrobial capacity, but also increases the oxidative potential. In this way, a high NaOCl concentration can compromise dentine chemical properties and promote high aggression to periapical tissues [44]. In this scenario, NTPP could be helpful for not significantly damaging the microhardness and roughness of dentine [28].

Aiming to demonstrate the efficacy of NTPP in realistic conditions, Simoncelli et al. [17] tested different protocols of root canal irrigation with PAW and the direct NTPP exposure using a plasma pistol. The results indicated that the exposure to PAW is enough to promote a significant reduction in the microbial load after 1 min of irrigation. Otherwise, the best results in bacteria inactivation were obtained in a dry environment and with direct NTPP exposure. These findings corroborate the results of Fridman et al. [15], which also verified the most effective bacterial inactivation on bacterial colonies from the skin after direct treatment. The authors attributed this effectiveness in sterilization to charged particles, microtherm effects, and vacuum UV radiation that were generated at the bacterial surface.

In view of the mentioned studies with microorganisms of endodontic interest, the use of NTPP has been demonstrated to be an effective adjunct to conventional treatments, although the best exposure parameters, considering the particularities of root canals, still need to be studied. It is important to emphasize that besides additional in vitro studies to set up the best parameters, clinical studies are extremely important to verify this effectiveness in vivo. The lack of clinical studies in the literature, and consequently in this review, represents a limitation of the present work.

Few in vitro studies have been conducted with PAW in the endodontic field, but they also showed promising findings with good perspectives to be used in the area. A possible advantage of using PAW would be that it may keep the antimicrobial effect for over one month after its generation, when stored at at least −80 °C [45,46]. In this way, the dentists would not necessarily need to have the plasma device in the clinic, which can be a limitation of using NTPP in dental offices. Moreover, the liquid form from PAW could be a solution to root canal anatomy complexity, which can also be pointed out as a limitation of NTPP. Additionally, treatment with PAW would probably be more comfortable for the patient than NTPP, depending on the time of exposure of the direct treatment. Therefore, new in vitro and in vivo studies with PAW are welcome and required to establish its real usefulness in the endodontic area.

## 6. Conclusions

Keeping in mind that conventional methods and techniques have limitations which often lead to root canal reinfection, NTPP could be used for endodontic disinfection as an adjunct treatment to root canal disinfection. Its efficacy in eliminating endodontic pathogens in vitro and ex vivo has been successfully demonstrated, mainly against *E. faecalis* and *C. albicans.*

The combination of NTPP and conventional antimicrobial methods has shown, in general, higher antimicrobial activity than the isolated treatments. This association may be an important method for decreasing the plasma exposure time and enhancing antimicrobial effectiveness, since long exposure and root canal anatomy complexity can build limiting factors to direct plasma exposure as a single treatment in clinical practice.

Future studies, including clinical investigations, are needed to confirm the effectiveness of NTPP and to set up the best parameters in which this technology can contribute to endodontic disinfection.

## Figures and Tables

**Table 1 biomedicines-11-01401-t001:** Studies on the use of NTPP in Endodontics.

Author/ Year	Type of Study	Plasma Exposure Method	Microorganism	Working Gas	Exposure Time	Distance from Apparatus to the Substrate	Other Antimicrobial Agent	Main Findings
Jiyang et al. [31]	In vitro and ex vivo	Direct	*Bacillus atrophaeus*	He and He (1%)O_2_ He (1%)O_2_	1 min 3 min	5 mm	No	He/(1%)O_2_ NTPP is more antimicrobial than He-NTPP alone against *B. atrophaeus*
Pan et al. [3]	In vitro	Direct	*E. faecalis*	Ar/O_2_[2%]	2–10 min	5 mm	No	NTPP was effective against *E. faecalis.* The exposure time of 8 or 10 min had significantly higher antimicrobial efficacy
Bansode et al. [32]	In vitro	Direct	*E. faecalis*	He He/O_2_	2 min	2–3 cm	Chlorhexidine (CLX)	A significant reduction in the biofilm viability was observed after chlorohexidine or NTPP treatment
Du et al. [33]	In vitro	Direct	*E. faecalis and multispecies biofilms*	He and He/O_2_	2–5 min	5 mm	CLX	Modified nonequilibrium plasma was more effective in killing *E. faecalis* and multispecies biofilms at both 2 and 5 min than conventional plasma. No significant difference was detected between nonequilibrium plasma and CHX groups
Habib, Hottel and Hong [4]	Ex vivo	Direct	*E. faecalis*	Argon	2 min	Not indicated	No	NTPP presented significant antimicrobial effects. It was as effective as 6% sodium hypochlorite
Jablonowski et al. [34]	Ex vivo	Direct	*E. faecalis*	Argon	3 min	2 mm	NaOCl and CLX	All treatments led to significant reduction of *E. faecalis* compared to NaOCl. NTPP was the most effective treatment
Schaudinn et al. [27]	Ex vivo	Direct	*Biofilm*	He/O_2_	30 min	Not indicated	NaOCl	NTPP showed lower biofilm removal than 6% NaOCl
Üreyen Kaya et al. [35]	Ex vivo	Direct	*E. faecalis*	He/O_2_	2 min	1 mm	NaOCL, Ozone	The highest antibacterial activity was observed in the NaOCl, NTPP and ozone groups. In the middle third of the root canal wall NTPP presented superior efficacy than NaOCl
Doria et al. [36]	In vitro	Direct	*Candida albicans*	Argon and	10 min	30 mm	No	Cell viability tests indicated that only about 8% of the yeast cells treated with Argon+ compressed air plasma
				Argon+ compressed air				could survive, proliferate and/or generate other cells. This treatment was the most effective in *Candida albicans* biofilm inactivation
Herbst et al. [37]	Ex vivo	Direct	*E. faecalis*	Argon	30 s (CLX) 60 s (NTPP)	1 mm	CLX	The highest antimicrobial action was observed in the association of NTPP and CLX, followed by NTPP and CLX alone
Simoncelli et al. [17]	In vitro and model	Direct and indirect	*E. faecalis*	He	3 min (direct) 1 min (indirect)	5 mm	NaOCl and CLX	The highest level of bacterial inactivation was observed in the dry environment by direct exposure, but a relevant bacterial load reduction was also obtained when the root canal system was irrigated with PAW
Armand et al. [1]	Ex vivo	Direct	*E. faecalis*	He and He/(0.5%)O_2_	2–8 min	2 mm	PDT	All the modalities showed a significant reduction in bacteria after treatment, and He/O_2_ NTPP was the most effective against *E. faecalis*, followed by photodynamic therapy and He plasma, respectively
Ledernez et al. [18]	In vitro	Direct	*E. faecalis, Streptococcus mutans Staphylococcus aureus Pseudomonas aeruginosa Escherichia coli*	He/(1%)O_2_	3 min	1–3 mm	No	After treatment, the median size of the bacterium-free area in Petri dishes of E. faecalis was 0.25 mm^2^, which corresponded to five times the area of the plasma nozzle and matched the target surface area in root canal treatment. This value was even larger for other investigated bacteria. *E. coli* presented the largest median bacterium-free surface area (2.5 mm^2^)
Li et al. [31]	In vitro	Direct	*E. faecalis*	Ar/O_2_	3, 6, 9 and 12 min	10 mm	Ca(OH)_2_, 2% CLX gel and Ca(OH)_2_/CLX	There were no detectable live bacteria after 12 min of NTPP treatment
Sallewong et al. [24]	Ex vivo	Direct	*E. faecalis*	He/O_2_	1 min	Not indicated	NaOCl	NTPP showed significant bacterial reduction as well as NaOCl and NaOCl + NTPP. The NaOCl + NTPP group significantly reduced *E. faecalis* in the deeper dentin level compared to the other groups
Li et al. [31]	In vitro	Indirect	*E. faecalis*	Compressed air	10–90 s	20 mm	No	PAW treatment inhibited *E. faecalis* biofilm and decreased quorum-sensing-related virulence genes expression
Kerli-kowski et al. [23]	Ex vivo	Direct	*Candida albicans*	Ar/O_2_	6–12 min	1–2 mm	CLX NaOCL OCT	NTPP presented the highest disinfection efficiency among all the treatments, after 6 and 12 min of exposure

## Data Availability

The data presented in this study are available on request from the corresponding author. The data are not publicly available due to privacy and ethical restrictions.

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
