# Peer review of "Non-Thermal Atmospheric Pressure Plasma Application in Endodontics"

_biomedicines, 2023, doi:10.3390/biomedicines11051401_

Round 1
Reviewer 1 Report
Non-thermal atmospheric pressure plasma application in 2 Endodontics
· In general, the topic is interesting; however, the design (literature review) is uncertain. There was a lack of rigor in research methods, such as the lack of literature search strategies, literature selection following exclusion and inclusion criteria, and protocol following the guidelines for preferred reporting items for systematic reviews. The literature review does not follow a standardized methodology such as a systematic review under specific search criteria (PRISMA statement) and conformation or identification of a question in PICO format.
The sections included are descriptive only and lacks depth. The main contributions and limitations of the review are not highlighted. In general, the study only confirms the results of previous studies and the contribution or originality is not appreciated.
To make the study more interesting, useful and attractive for clinicians and researchers, I suggest to make a systematic review.
More details or possible explanation of the topic should be mentioned, including not only a brief description of similar works, but a rational biologic explanation.
It is advisable to clarify the limitations identified and especially the future proposed works and recommendations based on the information generated.
Author Response
Thanks for pointing this out. We have performed some modifications in the structure of the text (topics and subtopics) to clarify that this is an integrative literature review and not a systematic review. We have also included the type of review in the section “design of the study”. For this reason, we did not standardized the methodology as necessary in a sistematic review, what would considerably change the format of the text and demand a lot of time. However, we have included some search strategies to inform the reader how the study was conducted in this review.
The main contributions and limitations have also been added in the discussion and conclusion. Considering that only in vitro and ex vivo studies were included in this review, and the lack of clinical studies in the area, we can not recommend yet the use of NTPP in clinical practice. However, the need for future clinical studies was pointed out since promising in vitro and ex vivo findings have been demonstrated.
All the modifications are highlighted in the text.
Reviewer 2 Report
Dear authors,
thank you for your interesting review article. Please address the following points:
1. Please specify the inclusion and exclusion criteria more in detail.
2. Please discuss the limitations of non-thermal atmospheric pressure plasma application and also provide some information in the conclusion section. Please answer the question: “Can non-thermal atmospheric pressure plasma application be recommended for endodontic disinfection?” Please add further information upon this issue to the conclusions.
Author Response
Please specify the inclusion and exclusion criteria more in detail.
Response: Thank you for pointing this out. We have add more details in the section “design of the study”.
All the modifications are highlighted in the text.
Please discuss the limitations of non-thermal atmospheric pressure plasma application and also provide some information in the conclusion section. Please answer the question: “Can non-thermal atmospheric pressure plasma application be recommended for endodontic disinfection?” Please add further information upon this issue to the conclusions.
Response: Thank you for pointing this out. Important limitations of non-thermal atmospheric pressure plasma were discussed in the discussion section (last paragraph) and also mentioned in the conclusion. Considering that only in vitro and ex vivo studies were included in this review, and the lack of clinical studies in the area, we can not recommend yet the use of NTPP or PAW in the clinical practice. However, the need for future clinical studies was pointed in the text.
All the modifications are highlighted in the text.

Reviewer 3 Report
Good article I have no comments
Author Response
Thank you very much for the review
Round 2
Reviewer 1 Report
Unfortunately, the observations made previously were not attended to; Specifically, it was not addressed to carry out a systematic review with clear criteria for the evaluation and selection of the articles available and used in the review. The concept of integrative review is not clear as a design and it is not justified either. Therefore, the manuscript cannot be accepted in the form it is presented. Authors are once again suggested to analyse the observation and convert the manuscript to a systematic review and if they consider it, make a new submission.